# Unveiling TIMPs: A Systematic Review of Their Role as Biomarkers in Atherosclerosis and Coronary Artery Disease

**DOI:** 10.3390/diseases12080177

**Published:** 2024-08-02

**Authors:** Amilia Aminuddin, Nazirah Samah, Ubashini Vijakumaran, Nur Aishah Che Roos, Faridah Mohd Nor, Wan Mohammad Hafiz Wan Razali, Shawal Faizal Mohamad, Beh Boon Cong, Faizal Amri Hamzah, Adila A. Hamid, Azizah Ugusman

**Affiliations:** 1Department of Physiology, Faculty of Medicine, Universiti Kebangsaan Malaysia, Jalan Yaacob Latif, Bandar Tun Razak, Cheras, Kuala Lumpur 56000, Malaysia; p124809@siswa.ukm.edu.my (N.S.); adilahamid@ppukm.ukm.edu.my (A.A.H.); 2Department of Tissue Engineering and Regenerative Medicine, Faculty of Medicine, Universiti Kebangsaan Malaysia, Jalan Yaacob Latif, Bandar Tun Razak, Cheras, Kuala Lumpur 56000, Malaysia; ubashinivijakumaran@ukm.edu.my; 3Faculty of Medicine and Defence Health, National Defense University of Malaysia, Kem, Sungai Besi, Kuala Lumpur 57000, Malaysia; nuraishah@upnm.edu.my; 4Forensic Unit, Department of Pathology, Faculty of Medicine, Universiti Kebangsaan Malaysia, Jalan Yaacob Latif, Bandar Tun Razak, Cheras, Kuala Lumpur 56000, Malaysia; faridah.nor@ukm.edu.my (F.M.N.); drhafizrazali@gmail.com (W.M.H.W.R.); 5Department of Forensic Pathology, Faculty of Medicine, Sungai Buloh Campus, Universiti Teknologi MARA, Sungai Buloh 47000, Selangor, Malaysia; 6Cardiology Unit, Department of Internal Medicine, Hospital Canselor Tuanku Muhriz, Jalan Yaacob Latif, Bandar Tun Razak, Cheras, Kuala Lumpur 56000, Malaysia; drshawal81.hctm@ukm.edu.my (S.F.M.); dr.behbooncong@ppukm.ukm.edu.my (B.B.C.); 7Department of Emergency Medicine, Faculty of Medicine, Universiti Kebangsaan Malaysia, Jalan Yaacob Latif, Bandar Tun Razak, Cheras, Kuala Lumpur 56000, Malaysia; drfaizalamri@ukm.edu.my

**Keywords:** atherosclerosis, coronary artery disease, tissue inhibitor of metalloproteinases

## Abstract

Coronary artery disease (CAD) is the leading cause of death globally and is a heart condition involving insufficient blood supply to the heart muscle due to atherosclerotic plaque formation. Atherosclerosis is a chronic disease in which plaques, made up of fat, cholesterol, calcium, and other substances, build up on the inner walls of arteries. Recently, there has been growing interest in finding reliable biomarkers to understand the pathogenesis and progression of atherosclerosis. Tissue Inhibitors of Metalloproteinases (TIMPs) have emerged as potential candidates for monitoring atherosclerotic development. TIMPs are a family of endogenous proteins that regulate matrix metalloproteinases (MMPs), enzymes involved in remodeling the extracellular matrix. A systematic search using Prisma guidelines was conducted and eleven studies were selected from four different databases: Web of Science (WOS), Scopus, Ovid, and PubMed. The Newcastle–Ottawa Scale (NOS) score was used to assess the risk of bias for each study. A meta-analysis was performed, and the hazard ratio (HR) and its 95% confidence interval (CI) were determined. Among the eleven studies, six reported a positive association between higher levels of TIMPs and an increased risk of atherosclerosis. Conversely, four studies support low TIMPs with high CAD risk and one study showed no significant association between TIMP-2 G-418C polymorphism and CAD. This divergence in findings underscores the complexity of the relationship between TIMPs, atherosclerosis, and CAD. In addition, a meta-analysis from two studies yielded a HR (95% CI) of 1.42 (1.16–1.74; *p* < 0.001; *I*^2^ = 0%) for TIMP-2 in predicting major adverse cardiovascular events (MACEs). In conclusion, the existing evidence supports the notion that TIMPs can serve as biomarkers for predicting the severity of atherosclerosis, myocardial damage, and future MACEs among CAD patients. However, further exploration is warranted through larger-scale human studies, coupled with in vitro and in vivo investigations.

## 1. Introduction

Atherosclerosis, the accumulation of plaque in arterial walls, underlies various cardiovascular diseases (CVDs), including myocardial infarction, stroke, and heart failure [1]. It is a multifactorial vascular disease that stands as the leading cause of mortality and morbidity worldwide. The identification and modulation of biomarkers in the development of atherosclerosis could be pivotal for early, subclinical diagnosis and therapeutic interventions. The concept of biomarkers was introduced in 1980 [2], reflecting measurable indicators of some biological state or condition. They are often used to diagnose or monitor a disease, predict disease progression, or evaluate the response to a therapeutic intervention [3,4]. Ideally, biomarkers should exhibit high sensitivity, high repeatability of results, and be economically and logistically feasible for clinical application [5]. Often cardiovascular risks were evaluated with the use of classical biomarkers such as circulating plasma LDL-cholesterol, HDL-cholesterol, or serum triglycerides and high sensitivity C-reactive protein (hsCRP) or N-terminal pro-brain natriuretic peptide (NT-proBNP) [6]. Over time, the discovery of novel biomarkers through genomic and transcriptomic searches, as well as small metabolites identified via metabolomics and lipidomics, has become increasingly elucidated [7]. Markers such as interleukin 6 (IL-6), myeloperoxidase (MPO), matrix metalloproteinase 9 (MMP-9), and intercellular and vascular cell adhesion molecules are also used as biomarkers in the development of chronic atherosclerosis development [8,9].

Matrix metalloproteinases (MMPs), enzymes crucial in atherosclerosis pathogenesis, degrade extracellular matrix proteins like collagen and elastin [10], contributing to clinical outcomes such as myocardial infarction and stroke [11]. MMPs also play a vital role in maintaining vessel integrity and promoting angiogenesis [12]. A study by Scholtes et al. [13] underscored MMP-2’s significance in atherosclerosis and plaque instability. Analyzing carotid atherosclerotic plaque samples from patients undergoing carotid endarterectomy (CEA), the study revealed elevated MMP-2 expression levels. Notably, unstable plaques exhibited higher MMP-2 expression compared to stable ones [13]. Thus, MMPs serve as markers for plaque vulnerability, predicting complications such as plaque rupture and subsequent cardiovascular events.

MMP activity within atherosclerotic plaques is tightly regulated by endogenous tissue inhibitors of metalloproteinases (TIMPs), which serve to oppose matrix degradation [14]. TIMPs comprise a family of proteins, which are TIMP-1, TIMP-2, TIMP-3, and TIMP-4, each made up of 184–194 amino acids that are ≈21 kDa in molecular weight. Each TIMP exhibits varying degrees of specificity and selectivity towards different proteases, contributing to their diverse inhibitory profiles [15]. TIMPs are present in numerous cells within atherosclerotic plaques [16]. Endothelial and smooth muscle cells have been reported to express TIMP-2 and were decreased in plaque compared to healthy vessels [17]. Moreover, TIMP-3 is highly expressed [18,19,20] and downregulated [20] in heart tissue and in the blood of patients with ischemic cardiomyopathy [19,20]. Among the TIMPs, TIMP-3 stands out as it is bound to the extracellular matrix (ECM), whereas the others are predominantly localized in a soluble form within the ECM. This differential localization affects their mode of action [21]. They exert their inhibitory effect on MMPs through the reversible blockage, forming a 1:1 stoichiometric complex with MMPs [22]. This complex formation occurs through the amino-terminal domain of TIMPs, which obstructs the active site of MMPs, preventing them from degrading the extracellular matrix within atherosclerotic plaques. Beyond their role in MMP inhibition, TIMPs also influence various cellular processes within the plaque microenvironment. They contribute to the regulation of cell structure and function, as well as the production of cytokines, growth factors, and chemokines [22,23]. This intricate interplay between MMPs and TIMPs underscores their significance in modulating the dynamic balance of ECM turnover and remodeling within atherosclerotic lesions.

TIMPs have unique roles in the progression of atherosclerosis. During plaque rupture, MMP secretion is elevated while TIMPs are not, especially in the macrophage-rich area, which subsequently accelerates the proteolytic action [24]. The counterbalance of TIMP-1, TIMP-2, and TIMP-3 to the expression of MMP-2 and MMP-9 showed a beneficial role in the development of an aneurysm [25,26]. A study by Howard et al. (1991) found that TIMP-2 in the fibroblast has the highest specific activity to inactivate MMP-9 [27]. To conclude, findings showed that TIMPs were decreased while MMPs increased in atherosclerosis plaque [24,26]. Moreover, a case–control study elucidated the correlation between TIMP-2 polymorphism and large artery atherosclerotic (LAA) stroke in a southern Chinese Han population. They reported that rs4789936 polymorphism in the TIMP-2 gene was associated with a lower risk of LAA stroke [28]. Differential effects of TIMP-1 and 2 were also reported where TIMP-2 suppressed MMP-14-dependent monocyte/macrophage infiltration into the atherosclerotic lesion in mice models [26]. This finding underscores the nuanced roles of different TIMPs in modulating key processes involved in atherosclerosis progression.

Nevertheless, there is a notable imbalance in research attention, with a predominant focus on MMPs’ actions, often overshadowing the crucial role played by TIMPs in the context of atherosclerosis. As our comprehension of the intricate molecular mechanisms governing atherosclerosis advances, it becomes imperative to shift the spotlight toward investigating and understanding the significance of TIMPs. Recognizing and validating novel biomarkers, particularly those involving TIMPs, present a promising avenue for enhancing risk stratification, early diagnosis, and monitoring the effectiveness of therapeutic interventions. In response to this research gap, we conducted a systematic exploration to gain deeper insights into the specific role that TIMPs play in the complex landscape of atherosclerosis. This review will consolidate current evidence and discuss the implications of TIMPs as emerging biomarkers in the context of atherosclerosis, fostering a comprehensive understanding of their diagnostic and prognostic potential in cardiovascular health. This systematic review collectively presents the production and underlying effects of TIMP 1, 2, 3, and 4. In addition, understanding the complex regulatory mechanisms involving TIMPs provides valuable insights into the pathophysiology of atherosclerosis and highlights their potential as therapeutic targets for interventions aimed at stabilizing vulnerable plaques and reducing the risk of cardiovascular events.

## 2. Methodology

This review was conducted in accordance with the Preferred Reporting Items for Systematic Reviews and Meta-Analysis (PRISMA) guidelines [29]. The PRISMA checklist is reported in Appendix A and Appendix B. The review protocol has been registered with the International Platform of Registered Systematic Review Protocols (registration number INPLASY202470070) [30].

### 2.1. Research Question and Search Strategy

The research question was formulated using the ‘PICO’ framework. Patients with CAD were the ‘Population (P)’, the control was defined as a ‘Comparison (C)’, the measurement of TIMPs’ level/expression or the associations between TIMPs and specific parameters such as MACEs and myocardial damage were defined as ‘Outcome (O)’, and exposure for certain TIMP genes was defined as ’Intervention (I)’ (for related study). Hence, the formulated question was as follows: are TIMPs emerging biomarkers of atherosclerosis in CAD? Further, were TIMPs also associated with major cardiac events and heart damage? The literature search was carried out across four internet databases (Pubmed, Scopus, Web of Science, and Ovid), with inclusion criteria limited to studies published between 2014 and January 2024. The keywords used for the search were (TIMP OR “Tissue inhibitor of metalloproteinases”) AND (Atherosclerosis OR “Ischemic heart disease” OR “Cardiovascular disease” OR “Coronary artery disease” OR “Acute coronary syndrome” OR “Myocardial infarction”).

### 2.2. Study Criteria

Three researchers (UV, NS, and AA) independently reviewed each article, focusing on adherence to the specified inclusion and exclusion criteria. Inclusion criteria encompassed (1) full-text of peer-reviewed original articles written in English, (2) research examining TIMPs and their association with atherosclerosis, and (3) clinical studies involving adult patients diagnosed with CAD confirmed via coronary angiogram, regardless of gender. The exclusion criteria comprised (1) original articles not in English, (2) reviews, conference abstracts, editorials, newsletters, books, and book chapters, (3) in vitro research, and (4) studies involving animals.

### 2.3. Article Selection and Data Extraction

The article selection process involved three phases. Initially, articles were screened based on their titles and types, with review or editorial articles being disregarded. Subsequently, abstracts were scrutinized to eliminate irrelevant articles concerning TIMPs and atherosclerosis. Finally, the remaining articles underwent thorough full-text analysis, and those not meeting the inclusion criteria were excluded. For data extraction, three researchers (AA, UV, and NS) independently compiled information such as the study’s first author, design, subject characteristics including age and gender, methods of TIMP measurement, and the association between TIMPs and atherosclerosis.

### 2.4. Risk of Bias Assessment

Three reviewers (UV, NS, and AAH) conducted independent assessments of the selected articles’ risk of bias. The Newcastle–Ottawa Scale (NOS) was utilized to critically evaluate the quality of risk of bias for each of the articles [31]. In cohort and cross-sectional studies, the NOS evaluated the selection of study groups (exposed and non-exposed), their comparability, and the assessment of outcomes. While, for case–control studies, the NOS assessed the selection of study groups (cases and controls), their comparability, and the determination of exposure for both cases and controls. A total of eight items from the three domains could receive a star rating, with each item being awarded a minimum of one star or a maximum of two stars. Studies receiving a total score of seven to nine stars were considered high quality, those with four to six stars were considered fair quality, and those with one to three stars were considered low quality.

### 2.5. Statistical Analysis

A meta-analysis was performed using Review Manager (RevMan) 5.4 software (The Cochrane Collaboration 2020). The hazard ratio (HR) and its 95% confidence interval (CI) generated from the multivariate Cox proportional hazard analysis were used as the effect estimate in reporting the role of TIMPs as predictors of major adverse cardiovascular events (MACEs) among CAD patients. The heterogeneity between studies was evaluated using (1) the Chi-squared test with a ***p***-value of less than 0.10 to denote statistical significance and (2) the Higgin’s *I*^2^ statistic [32]. An *I*^2^ value of less than 25% was regarded as low heterogeneity, while an *I*^2^ value of 75% or more was considered as high heterogeneity. A fixed-effect (FE) model was used due to the small number of studies available for meta-analysis. A *p*-value less than 0.05 indicated statistical significance. To ensure consistency, models that were adjusted for similar confounding factors were pooled together. In the primary meta-analysis, Model 1 from Wang et al. was pooled together with Model 2 from Huang et al. because these models were adjusted for conventional CAD risk factors such as age, sex, and total cholesterol [33,34]. A sensitivity analysis was conducted by pooling models that were adjusted for additional confounding factors from both studies to assess the robustness of the effect estimate generated. No subgroup analysis was performed due to the limited number of studies. A publication bias analysis was not reported as fewer than 10 studies were included in the meta-analysis.

## 3. Results

### 3.1. Search Results

This review was reported according to the guidelines outlined by the Preferred Reporting Items for Systematic Reviews and Meta-Analysis (PRISMA). From the search, a total of 6974 articles were obtained from the four online databases, which are Pubmed (415), Scopus (3646), Web of Science (WOS) (425), and Ovid (2488). All the articles were pooled into Mendeley Desktop Version 1.19.8, and 1833 duplicates were removed. A total of 147 non-English articles were also removed before the screening of titles and abstracts. Screening of titles and abstracts to include articles investigating the association of TIMPs, atherosclerosis, and CAD excluded 4915 articles. These numbers also included studies that were reviews and editorials. The remaining 61 retrievable articles were read in full to determine satisfaction with the eligibility criteria. As a result, only 11 studies were selected to be included in this review. The flow chart in Figure 1 summarizes the study selection approach.

### 3.2. Quality Evaluation

Quality assessment using NOS for all the selected 11 articles was tabulated in Table 1 and Table 2. It was revealed that the score ranges of the studies reviewed were between 5 and 8 (fair to high quality). In detail, eight studies fell into high-quality [33,34,35,36,37,38,39,40]. Meanwhile, three studies were classified as fair quality [39,41,42].

### 3.3. General Characteristics of the Included Studies

The 11 studies included in this study were published between 2014 and January 2024. Moreover, there were 36.4% (n = 4) studies that were cohort studies, 36.4% (n = 4) case–control studies, and 27.2% (n = 3) cross-sectional studies. CAD including acute coronary syndrome (ACS) comprising STEMI, NSTEMI, MI, and unstable angina was studied in four studies (36.4%). While CAD by coronary angiography was studied in five studies (45.4%), CAD underwent bypass in one study (9.1%), and atherosclerotic coronary plaque was studied in one study (9.1%). The majority of the subjects were male and ranged from middle to old age. Most of the studies assessed TIMP levels across various types of samples, including serum [33,35,42,43], plasma [34,40], and blood [36,39]. Gene expression in circulating leukocytes [41] or PBMCs [37] and post-mortem coronary arteries [41] were also studied. Moreover, one piece of research studied TIMP-1 gene polymorphism from blood DNA [38]. Among these 11 studies reviewed, TIMP 1 was the most studied compared to other types of TIMPs (n = 6, 54%).

There were also several outcomes investigated. Cohort studies looked into the predictive value of circulating TIMPs for MACEs [33,34,35] or cardiac remodeling or infarct size [36]. For observational studies, different outcomes were studied. TIMPs were suggested as markers of myocardial damage [39], CAD markers [37,40,43], or atherosclerotic markers [41,42] based on the method used. In addition, one study investigated the association between TIMP-1 polymorphism from whole blood with CAD [38]. Myocardial damage was assessed by the level of troponin T or infarct size via MRI and Spect [36,39]. The CAD marker was suggested by looking at the level of circulating TIMP [40,43] or transcriptional activity of TIMPs [37] among CAD patients. For atherosclerotic markers, TIMP expression in the coronary artery plaque [41] or among CAD patients with fibroatheroma plaque [42] was studied.

### 3.4. Finding Summary

Elaborated data extraction is given in Table 3 and Table 4. Overall from the 11 studies that have been selected, only four studies showed TIMP downregulation associated with a higher risk of atherosclerosis [37,40,41,42] while six studies [33,34,35,36,39,43] reported higher expression of TIMPs closely related to atherosclerosis manifestation. One study reported no significant association between TIMP-2 G-418C polymorphism and CAD [38].

Meta-analysis was conducted to determine the predictive value of TIMP-1 for MACEs. From four prospective studies [33,34,35,36], only two studies provided suitable data to be pooled together [33,34]. These two studies involved a similar cohort of patients at baseline, which were CAD patients, and studied the predictive value of TIMP-1 for MACEs. The remaining two studies were not suitable because they involved healthy subjects at baseline [35] and the predictive value for MACEs was not available [36]. The meta-analysis showed that the HR (95% CI) was 1.42 (1.16–1.74; *p* < 0.001; *I*^2^ = 0%), suggesting that TIMP-1 was significantly associated with an increased hazard of MACEs (Figure 2). No heterogeneity was observed. A sensitivity analysis was conducted by including Model 2 in [34] and Model 3 in [33], which obtained the HR (95% CI) of 1.69 (1.36–2.1; *p* < 0.001; *I*^2^ = 0%) (Figure 3).

## 4. Discussion

This review was carried out to explore the emerging role of TIMPs as biomarkers in atherosclerosis, shedding light on their intricate involvement in vascular remodeling, inflammation, and plaque stability. The delicate balance between MMPs and TIMPs is essential for maintaining vascular homeostasis, and disruptions in this equilibrium have been implicated in various cardiovascular diseases, including atherosclerosis. Several studies have suggested that alterations in the expression levels of TIMPs, particularly within the atherosclerotic lesion microenvironment, may indicate disease severity and progression. Collectively, we have analyzed the studies that reported the presence of TIMP-1–4 in atherosclerosis and control subjects. From the systematic search we have carried out, few studies found that TIMP-1 and 2 significantly decreased in coronary artery disease (CAD) patients compared to control, while they increased in control subjects. They also showed an inverse correlation with MMPs [40].

Research spanning over two decades has extensively explored the connection between matrix metalloproteinases (MMPs) and atherosclerotic plaque. Notably, MMPs have been identified in vulnerable regions of plaques, and regions rich in MMPs are more susceptible to rupture [43,44,45,46]. In this context, the involvement of tissue inhibitors of metalloproteinases (TIMPs) becomes crucial. Their role is to counteract the elevated MMP levels and maintain a balance in the extracellular matrix. When TIMPs fail in this task, it leads to accelerated atherosclerosis, as reported by Ben et al. [40]. Specifically, the inhibition of MMP-9 by TIMP-1 and TIMP-2 has been identified as a mechanism that halts plaque progression and instability by altering foam cell behavior [22]. This emphasizes the importance of understanding the intricate interplay between MMPs, and TIMPs, and their impact on atherosclerosis for developing effective therapeutic strategies.

In a study conducted by Ezhov et al. [42], it was observed that TIMP-2 levels were lower in fibroatheroma, a condition associated with increased cardiovascular risks, particularly in diabetic patients, when compared to lipid plaques [42]. Fibroatheroma is recognized as a primary contributor to coronary fatality while thin cap atheroma, identified as a precursor to plaque rupture, demonstrated a concurrent increase in MMP-7 and MMP-9 levels. This implies that a reduced level of TIMP-2 may fail to effectively inhibit the actions of MMPs, thereby contributing to the formation of fibroatheroma. Besides this, TIMP-2 showed the greatest protective role compared to TIMP-1 in reducing smooth muscle cell proliferation and the necrotic core by suppressing MMP-14-dependent monocyte/macrophage accumulation into plaques [26].

Together with this, the action of TIMP-3 in vascular remodeling has been identified as a checkpoint that regulates angiogenesis, extracellular matrix remodeling, and metabolic inflammatory signaling [47]. It is unique among four TIMPs as it is the only one that has the widest range of substrates, including all MMPs, several ADAMs (disintegrin and metalloproteinases), ADAMTSs (ADAM with thrombospondin motifs), and strong affinity for the proteoglycans in the extracellular matrix [22,48,49]. TIMP-3 also plays a therapeutic function in maintaining tissue integrity by acting as a protective factor against myocardial infarction, ischemia-associated cardiomyopathy, and heart fibrosis [50]. TIMP-3 induces smooth muscle cell proliferation and migration by suppressing the activity of MMP-2, MMP-9, and TNF-α secretion in the development of an atherosclerotic abdominal artery aneurysm (AAA) [51].

Furthermore, the absence of TIMP-3 has been demonstrated to polarize macrophages and intensify inflammation in atherosclerosis, as evidenced in ApoE null mice [52]. Notably, studies involving intracoronary delivery of recombinant TIMP-3 have shown promising outcomes in post-myocardial infarction (MI) remodeling, leading to reduced left ventricular (LV) dilation and lower levels of plasma NH2-terminal pro-brain natriuretic peptide, an established marker of heart failure progression [53]. Despite the scarcity of studies on TIMP-3 expression in living and autopsy samples, available research indicates no significant differences in TIMP-3 expression in circulating leukocytes and peri-coronary epicardial adipose tissues (EAT). However, a distinctive finding emerges from autopsy samples, where TIMP-3 exhibits a significant reduction in advanced atherosclerotic plaques compared to normal arterial segments [41]. This decline in TIMP-3 expression in advanced atheroma may be attributed to the abundance of macrophages within these lesions, potentially influencing TIMP-3 gene expression. The implications of reduced TIMP-3 expression extend beyond its role in counteracting matrix metalloproteinases (MMPs) alone. It is associated with the proliferation of foam cell macrophages (FCMs), a pivotal process in accelerating plaque development [54,55]. Moreover, the coexistence of decreased TIMP-3 levels and heightened MMP expression is predominantly observed in intimal macrophages, as demonstrated in studies by Johnson et al. (2008) [56]. Consequently, the multifaceted role of TIMP-3 in atherosclerosis development suggests its involvement in modulating macrophage proliferation, emphasizing its significance beyond its conventional role in MMP inhibition.

In contrast, from the systematic search, six studies highlight the intricate and contrasting roles of Tissue Inhibitors of Metalloproteinases (TIMPs) in vascular remodeling, particularly in the context of coronary artery disease (CAD) and myocardial infarction (MI). Notably, upregulation of TIMP in CAD patients has been observed, suggesting a potential compensatory mechanism in response to elevated Matrix Metalloproteinase-9 (MMP-9) levels. In a randomized control trial involving 243 ST-segment Elevation Myocardial Infarction (STEMI) patients, it was found that the circulating MMP-9/TIMP-1 ratio was higher in the subacute phase compared to the stable phase [36]. This implies a dynamic regulation of these proteins during different stages of the disease. Elevated TIMP-1 levels in the subacute phase may be a response to counteract the increased MMP-9, indicating a potential role in tissue repair or remodeling. Moreover, the same study reported that the blood TIMP-1 correlated with larger infarct size and impaired LV function after STEMI.

Moreover, other studies have reported higher levels of serum TIMP-2 in patients with angiographically confirmed coronary artery stenosis compared to those without stenosis [43]. MMP-9 and TIMP-2 were also found to be elevated in plasma patients with premature coronary atherosclerosis [57], emphasizing the significance of these biomarkers in assessing coronary vascular health. A biomarker analysis from 389 male patients undergoing coronary angiography identified high plasma TIMP-1 as an independent predictor of myocardial infarction (MI) [58]. Furthermore, a 12-year follow-up study revealed that high blood TIMP-1 levels were independently associated with the long-term outcome of coronary bypass graft (CABG), suggesting a potential role in atherosclerotic progression [33]. Their study represents a novel finding, demonstrating the correlation between TIMP-1 and long-term clinical outcomes in CABG patients. The suggested TIMP-1 could modulate ventricular remodeling and subsequent heart failure to impact the long-term patency of CABG. Another study by Wang et al. highlighted a unique aspect where circulating TIMP-1, rather than MMP-9, independently predicted major adverse cardiovascular event (MACE) risk. In this study involving Chinese patients with myocardial infarction and a coronary arteriosclerosis group, elevated levels of both circulating MMP-9 and TIMP-1 were observed. However, the correlation with MACE risk was specifically attributed to TIMP-1 [34]. Other studies in the current review also have identified the role of TIMP-1 as a predictor of MACEs [35]. In a notable 13-year follow-up study conducted by Kormi et al., [35] the findings showed that serum TIMP-1 levels were a prominent contributor to cardiovascular disease (CVD) and CAD events, boasting the largest hazard ratio, even surpassing C-reactive protein (CRP). Similarly, the association between higher TIMP-1 levels and increased risk of MI and mortality, as demonstrated by Cavusoglu et al. [58] in a 2-year follow-up study, emphasizes the potential of TIMP-1 as a valuable prognostic marker. Lubos et al. [59] further supported these observations by revealing elevated TIMP-1 levels in patients with suspected coronary artery disease (CAD) who experienced fatal outcomes during follow-up. Notably, TIMP-1, expressed by a variety of cells including smooth muscle cells, endothelial cells, fibroblasts, cardiac myocytes, and macrophages, plays a crucial role in maintaining physiological processes. It exerts regulatory control over the differentiation and proliferation of cells, particularly in response to leukocyte influx during tissue injury [60]. This multifaceted involvement in cellular processes may elucidate the elevated levels of circulating TIMP-1 observed in association with various CVD endpoints. Supporting evidence from a Framingham Study involving 922 participants underscores the significance of TIMP-1 in cardiovascular health. Higher levels of TIMP-1 were found to be correlated with increased mortality and incidence of CAD, further emphasizing its pivotal role in cardiovascular outcomes [61].

From two studies by Huang et al. [33] and Wang et al. [34], our meta-analysis supported the role of TIMPs as a predictor of MACEs with a HR (95% CI) of 1.42 (1.16–1.74; *p* < 0.001; *I*^2^ = 0%). However, there were some differences between both studies, which could contribute to the robustness of the results. Firstly, although both models adjusted for similar conventional risk factors such as age, sex, and total cholesterol, other confounders were not similar between studies. For example, Model 2 by Huang et al. [33] was additionally adjusted for the manifestation of acute coronary syndrome, estimated glomerular filtration rate, left ventricular ejection fraction, and triglycerides, while Model 1 by Wang et al. [34] was adjusted for BMI, systolic blood pressure, diabetes, and smoking status. Secondly, the cohort patients of both studies were not identically similar. Although both studies involved CAD patients, patients in the work by Huang et al. [33] were at an advanced stage since the patients underwent CABG and received different types of treatment, compared to mild and moderate CAD in the work by Wang et al. [33]. Thirdly, patients in the work by Huang et al. [23] were much older. Fourthly, the use of patient medication may not be similar between studies due to the subjects’ age, the severity of the disease, and the advancement of treatment. Lastly, the definition of MACEs was quite different between both studies. In Wang et al. [34]. MACEs were defined as a composite of all-cause death, nonfatal acute myocardial infarction, revascularization (except for myocardial infarction), and readmission due to refractory angina pectoris (without revascularization), while the study by Huang et al. [33] defined MACEs as non-fatal myocardial infarction, non-fatal stroke, and cardiovascular.

One study determines the TIPM-2 gene polymorphism among the Turkish population [38]. The polymorphism has been studied in other diseases such as periodontitis [62] and chronic obstructive pulmonary disease [63]. The main finding in [38] showed that TIMP-2 G-418C polymorphism does not appear to be associated with CAD or MI in the studied Turkish population. However, the sample size was relatively small for a genetic study. Further research should focus on a larger sample size and explore other potential genetic contributors to CAD and MI.

While the traditional view implicates TIMP-1 in inhibiting matrix metalloproteinases (MMPs) to regulate the extracellular matrix (ECM) breakdown in thrombi, emerging evidence suggests a more nuanced role for TIMP-1. Beyond its mere reflection of a secondary response to MMP activity, the elevated TIMP-1 levels observed at the early stage of STEMI and just before PCI may signify a direct involvement in cardiac remodeling. It appears that TIMP-1 could play multiple roles, extending beyond MMP regulation, and potentially influencing broader processes associated with cardiac remodeling. When considering the amalgamation of the chosen studies, the discernible high heterogeneity becomes apparent, encompassing diverse study types, sample variations, disease cohorts, and follow-up durations. The divergent roles attributed to Tissue Inhibitors of Metalloproteinases (TIMPs) in predicting coronary artery disease (CAD) events may stem from an inadequately controlled population to cardiovascular disease (CVD) risk factors. To address this, future research could benefit from prolonged cohorts that include multi-ethnic populations. Additionally, the limited reporting of the MMPs/TIMPs ratio in a subset of studies underscores the importance of incorporating this ratio into future investigations. A comprehensive analysis of this ratio could provide valuable insights into the counteractive actions of TIMPs against MMPs. Therefore, the inclusion of this ratio in forthcoming studies is crucial for a more nuanced understanding of the intricate dynamics within the context of cardiovascular health.

## 5. Conclusions

This review highlights the multifaceted role of TIMPs in atherosclerosis and CAD, where both excessive and insufficient TIMP activity can be detrimental. Several studies reveal positive associations between higher levels of TIMPs and increased progression of atherosclerosis. While, in contrast, some studies reported low TIMP levels associated with an upregulated risk of coronary artery disease. In addition, meta-analysis suggests TIMP-2 can be a significant predictor of adverse cardiovascular outcomes, further emphasizing its importance in the pathophysiology of CAD. In conclusion, the relationship between TIMPs, atherosclerosis, and CAD is complex, with both high and low levels of TIMPs potentially contributing to disease risk. Further research is needed to elucidate the precise mechanisms and establish standardized methodologies to enable a more comprehensive understanding of the diagnostic and therapeutic implications of TIMPs in atherosclerosis.

## Figures and Tables

**Figure 1 diseases-12-00177-f001:**
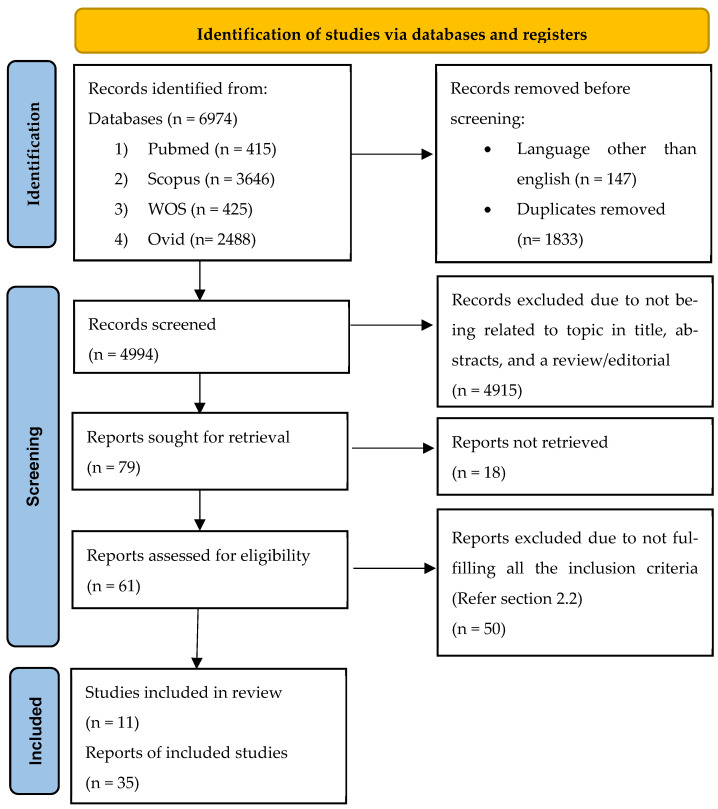
The flow chart illustrates the process of selecting and screening based on PRISMA guidelines.

**Figure 2 diseases-12-00177-f002:**
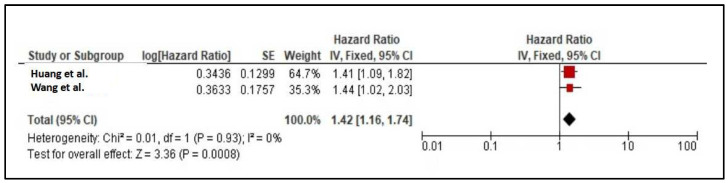
Meta-analysis of TIMP-1 as predictor for MACEs [33,34].

**Figure 3 diseases-12-00177-f003:**
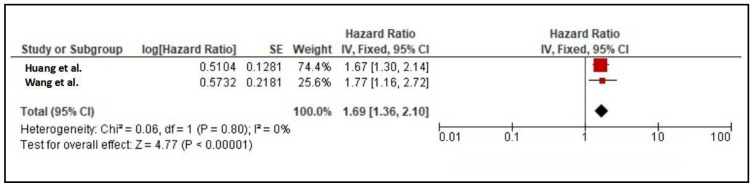
Sensitivity analysis: meta-analysis of TIMP-1 as a predictor for MACEs [33,34].

**Table 1 diseases-12-00177-t001:** The Newcastle–Ottawa Scale (NOS) for assessment of cohort/cross-sectional.

Author	Year	Type of Study	Selection				Comparability	Outcome			Total Score
			Representatives of the Exposed Cohort	Selection of the Non-Exposed Cohort	Ascertainment of Exposure	Demonstration that Outcome of Interest Was Not Present at Start of Study	Comparability of Cohorts on the Basis of the Design or Analysis	Assessment of Outcome	Was Follow-Up Long Enough for Outcomes to Occur	Adequacy of Follow-Up of Cohorts	
[33]	2023	Cohort	✵	✵	✵	✵	✵	✵	✵	✵	8
[39]	2022	Cross-sec	✵		✵	✵	✵	✵			5
[41]	2021	Cross-sec	✵	✵	✵	✵	✵	✵			6
[42]	2019	Cross-sec	✵		✵	✵	✵	✵			5
[36]	2018	Cohort	✵		✵	✵	✵	✵	✵	✵	7
[35]	2017	Cohort	✵	✵	✵	✵	✵	✵	✵	✵	8
[34]	2015	Cohort	✵	✵	✵	✵	✵	✵	✵	✵	8

✵ Indicates one star.

**Table 2 diseases-12-00177-t002:** The Newcastle–Ottawa Scale (NOS) for assessment of case–control.

Author	Year	Type of Study	Selection				Comparability	Exposure			Total Score
			Is the Case Definition Adequate?	Representativeness of the Cases	Selection of Controls	Definition of Controls	Comparability of Cases and Controls on the Basis of the Design or Analysis	Ascertainment of Exposure	Same Method of Ascertainment for Cases and Controls	Non-Response Rate	
[40]	2021	Case–control	✵	✵	✵	✵	✵	✵	✵	✵	8
[43]	2016	Case–control	✵	✵	✵	✵	✵	✵	✵	✵	8
[37]	2023	Case–control	✵	✵	✵	✵	✵	✵	✵	✵	8
[38]	2017	Case–control	✵	✵	✵	✵	✵	✵	✵	✵	8

✵ Indicates one star.

**Table 3 diseases-12-00177-t003:** Summary of data extraction table.

Reference	Type of Study/Trial Design	Subject Number and Characteristics				Sex	Mean Age (Years Old)	Sample Collection/Measurement Analysis
[33]	Prospective cohort- Follow up-12 years	CAD patients, underwent CABG (n = 234)				84.6% men	Men = 70.4 ± 10.5	Serum and plasma samples were collected after overnight fasting before the procedure
[34]	Prospective study/Cohort-Follow up-1, 3, 6, 12, 24, 36, 48, 60, 72, and 84 months after CAG	Patients with mild to moderate CAD (n = 522) Chinese population Hypertension treatment: 61.4% Statin medication: 52.7% Aspirin medication: 63.3%				67.7% men	60 (IQR: 52.4–66.9)	Fasting venous blood collected after CAG
[35]	Cohort Prospective study—follow-up for 13 years	7928 free from CVD at baseline Subjects divided into no CVD, CAD, MI, stroke, and death following the endpoint Hypertensive medication: ▪ No CVD, n = 840 (11.7%) ▪ CVD, n = 109 (14.3%) ▪ CAD, n = 78 (14.8%) ▪ MI, n = 52 (18.5%) ▪ Stroke, n = 41 (13.9%) ▪ Death, n = 110 (13.3%)				At follow up: ▪ No CVD, (3345; 46.7% men) ▪ CVD, (536; 70.3% men) ▪ CAD, (387; 73.3% men) ▪ MI, (205; 73% men) ▪ Stroke, (198; 67.1% men) ▪ Death (554; 67.1% male)	At follow-up: ▪ No CVD (46.7 ± 12.9) ▪ CVD (61.1 ± 9.1) ▪ CAD (61.1 ± 8.7 ▪ MI (61.8 ± 8.5) ▪ Stroke (61.9 ± 9.4)	Blood withdrawn after at least four hours fasting
[36]	Prospective/Cohort—follow-up at 3 days and 3 months post-AMI Multi-center randomized, controlled	n = 243 Norwegian STEMI patients Medications: ▪ ASA, n = 237 (98%) ▪ Clopidogrel, n = 23 (92%) ▪ Statins, n = 240 (99%) ▪ β-Blockers, n = 219 (91%) ▪ ACE inhibitors, n = 108 (45%) ▪ ARBS, n = 18 (7%)				61% women	60 (53, 67-IQR)	Fasting blood drawn at 8 and 10 a.m. on day 3 and 3 months post-AMI
[37]	Case–control study	Patients with CAD without heart failure (CAD, n = 40) Patients with CAD and heart failure (CADHF, n = 80) Healthy controls (HG, n = 30)				CAD = 97.5% men CADHF = 98.75% men HG = 50% men	CAD = 64.67 ± 10.45 CADHF = 67.35 ± 7.67 HG = 62.77 ± 9.29	Peripheral blood mononuclear cells were obtained from whole blood
[38]	Case–control study	Turkish population 122 CAD patients and 132 controls characterized by angiogram. CAD further divided into CAD (n = 84) and MI (n = 38)				CAD = 54.76% men MI = 84.2% Control = 45.5%	CAD = 60.40 ± 10.8 MI = 60.81 ± 12.4 Control = 57.56± 10.7	DNA was extracted from whole blood
[39]	Cross-sectional study	STEMI patients (n = 33) Treated with primary PCI and thrombus aspiration Medication: ▪ ASA = 6 (18%) ▪ Clopidogrel/Prasugrel/Ticagrelor = 2 (6%) ▪ Warfarin = 1 (3%) ▪ NOAC = 2 (6%) ▪ Beta blocker = 4 (12%) ▪ ACE-I = 0 (0%) ▪ AT-II-blocker = 5 (15%) ▪ Statins = 6 (18%) ▪ Diuretics = 4 (12%) ▪ Aldosterone antagonists = 0 (0%)				9% women 91% men	Women = 58.0	Blood sample: circulating TIMP and leucocyte (for RNA extraction) Coronary thrombi obtained during PCI: level of TIMP-1, TIMP-2, and RNA extraction
[40]	Case–control multi-centric study	CAD patients (n = 472) ▪ STEMI (n = 162) ▪ NSTEMI (n = 242) ▪ Stable angina (n = 67) Healthy controls (n = 285) Medication:				CAD = 67% men 33% women Control 66% men 34.4% women	CAD = (57.1 ± 11.4) Control = (55.01 ± 13.82)	Blood samples collected before coronary angiography for plasma TIMPs
			STEMI	NSTEMI	Stable angina			
		RAASI (%)	79	118	35			
		Insulin (%)	70	104	29			
		Lipid lowering agent (%)	87	133	36			
[41]	Cross-sectional study	(A) Living cases (n = 69) undergone invasive coronary angiography for CAD diagnosis and surgical pre-surgical assessment. Gensini and syntax scores were calculated (cut-off value of 8 for both scores) and divided into the following: Group with high plaque score (n = 48), defined as stenosis >50% in one of the artery. Five from this group underwent heart valve replacement and twenty underwent coronary artery bypass surgery (CABG) Group with low plaque score (n = 21). Nine of them underwent heart valve replacement (B) Post-mortem study case (n = 30) procured from male cadavers within 24 h after death CHD subjects (n = 26), non-cardiac trauma (n = 4)				High plaque, 64% men Low plaque, 50% men	High plaque (57.7 ± 10.91) Low plaque (57.93 ± 10.21) CHD-P (51.3 ± 12.1) CHD-nonP (51.3 ± 12.1) T-nonP (29.3 ± 16.3)	From living subject: (A) Collection of blood for leukocytes extraction from all patients (n = 69) (B) Collection of peri-coronary EATs from those undergone heart valve replacement and CABG: High plaque score (n = 25), low plaque score (n = 9) From post-mortem subject: (A) Collection of coronary artery ▪ Arteries with advanced atherosclerotic plaque (CHD-P) (n = 26) ▪ No plaque arterial segment from CHD subjects (CHD-nonP) (n = 25) ▪ Non-atherosclerotic coronary arteries (T-nonP) (n = 4) (B) Collection of peri-coronary epicardial adipose tissues (EATs) ▪ Surrounding arteries with advanced atherosclerotic plaque (CHD-P) (n = 10) ▪ Surrounding around arterial segment with no plaque from CHD subjects (CHD-nonP) (n = 10) ▪ Surrounding non-atherosclerotic coronary arteries (T-nonP) (n = 2)
[42]	Cross-sectional	Stable CHD (n = 32), underwent intravascular ultrasound (IVUS) Medication: Acetylsalicylic acid, n = 30 (94%) Clopidogrel, n = 13 (37.5%) Angiotensin-converting enzyme inhibitor, n = 18 (56%) Angiotensin receptor antagonists, n = 7 (26%) Beta blockers, n = 27 (84%) Calcium channel blockers, n = 11 (34%) Nitrates, n = 13 (41%) Atorvastatin, n = 32 (100%)				62.5% men	56.1 ± 8.0	Blood samples were drawn immediately prior to angiography for plasma TIMP-1 and TIMP-2
[43]	Case–controlSingle center	Those underwent coronary angiography (n = 70) Normal (n = 13) 1-vessel disease (n = 22) 2-vessel disease (n = 17) 3-vessel disease (n = 18)				57 men 13 women	CAD = 58.5 ± 10.2 Normal = 52.3 ± 10.0	Blood taken before angiography

**Table 4 diseases-12-00177-t004:** Summary of data extraction table.

Author	Methods/Measurement Analysis	Findings	Conclusion
[33]	Blood TIMP-1: ELISA	Baseline TIMP-1 value for all the subjects: 127.9 ± 63.67 ng/mL End point: MACEs (non-fatal myocardial infarction, non-fatal stroke, and cardiovascular death) In multivariable Cox regression analysis, 3 models had been developed for TIMP-1, HR (95% CI) (A) Model 1, adjusted for age and sex: 1.546 (1.322–1.809) (B) Model 2, adjusted for age, sex, manifest ACS, eGFR, left ventricular ejection fraction, TC, and triglycerides: 1.41 (1.093–1.819) (C) Model 3, adjusted for age, sex, manifest acute coronary syndrome, estimated glomerular filtration rate, left ventricular ejection fraction, completed revascularization or not: 1.666 (1.296–2.142)	High circulating TIMP-1 is a potential independent predictor of future MACE and mortality among CAD patients
[34]	Evaluation of lesion: coronary angiography Measurement of plasma TIMP-1: protein array	Baseline circulating TIMP-1 = 37 552.8 (24 369.7–52 216.5) pg/mL. MI, unstable and stable angina patients have high levels of MMP-9 and TIMP-1 MMP-9 has no correlation with outcome of with mild to moderate coronary artery lesions Endpoint: MACEs [composite of all-cause death, nonfatal acute myocardial infarction, revascularization (except for myocardial infarction), and readmission due to refractory angina pectoris (without revascularization)]. To estimate the risk of cardiovascular events, various Cox regression models were performed In Model 1 (adjusted for age, sex, BMI, systolic blood pressure, diabetes, smoking status, and TC), the HR (95% CI) for TIMP-1 was 1.438 (1.019–2.028, *p* = 0.039) In Model 2 [additional adjustment for therapeutic variables (hypertension, statin and aspirin treatments)] TIMP-1 showed a more statistically significant relationship with the outcome [1.774, (1.157–2.721), *p* = 0.009] In Model 3 [additional adjustment for discharge diagnosis, 1.608 (1.043–2.478), *p* = 0.031]	TIMP-1 is a potential candidate to predict long-term prognosis in patients with mild to moderate coronary artery lesions in a Chinese population
[35]	Measurement of baseline serum of TIMP-1 by chemiluminescent micro particle immunoassay (CMIA)	After 13 years, 89.6% survived without registered endpoints, 762 develop CVD (9.6%), which consist of CAD (n = 534, 6.7%), MI (n = 281, 3.5%), and stroke (n = 295, 3.7%), while 826 subjects (10.4%) died Serum TIMP-1 concentrations associated significantly with increased risk for all studied endpoints [adjusted HR (95% CI) per SD] CAD: 1.15 (1.05–1.26), MI: 1.26 (1.11–1.41), death 1.37 (1.27–1.49)	High-serum TIMP-1 is associated with increased risk for future CVD events and death among health population
[36]	Detection of blood TIMP-1, Troponin T -ELISA Myocardial perfusion imaging: SPECT Infarct size- Cardiac MRI	Day 3 levels of circulating TIMP-1 [193(171,215) ng/mL] was significantly correlated to infarct size assessed by troponin T (measured on day 3), SPECT and MRI (total and relative infarct volume) measured at 3 months (r = 0.159, 0.140, 0.234, and 0.227, respectively, p < 0.03 for all) In logistic regression, the highest quartile of TIMP-1 levels on day 3 showed an adjusted odds ratio of 5.0 (95% confidence interval [CI] 1.2–20.6) (*p* = 0.03) and 2.5 (95% = CI 1.0–5.9) (*p* = 0.04) for a large infarct size (>16.3% MRI-based and >21.8% SPECT-based, respectively) compared to quartile 1 MMP-9/TIMP-1 ratio was higher in the subacute phase (Day 3, 3.7) compared to the stable phase (3 months, 2.1) (p < 0.001)	Higher TIMP-1 has association with infarct size, supports TIMP-1 role in cardiac remodeling
[37]	Transcriptional activity of TIMP-1 gene in PBMCs: QRT-PCR technique	Transcriptional activity of the tissue metalloproteinase inhibitor 1 (TIMP-1) gene was significantly higher in the group of patients with CAD without HF and in patients with CAD and heart failure compared to healthy With the advancement of heart failure (decrease in LVEF), a decrease in the transcriptional activity of TIMP-1 gene was found	TIMP-1 gene transcriptional activity was significantly lower among CAD and negatively correlated with the severity of heart failure, which make it as useful diagnostic and prognostic markers in clinical practice
[38]	TIMP-2 gene polymorphism: PCR	No significant differences were found between TIMP-2 G-418C polymorphism and CAD or MI in the population	TIMP-2 G-418C polymorphism was not associated with increased or decreased risk of CAD among the Turkish population
[39]	Measurement of circulating TIMP-1, TIMP-2: ELISA Expression of TIMP-1 and TIMP-2 in thrombi and leucocyte: RT PCR Level of TIMP in thrombi: immunostaining	Expression of TIMP-1 in thrombi and in leukocytes correlated significantly to peak troponin T (rho = 0.393 *p* = 0.026, rho = 0.469 *p* = 0.006, respectively). Circulating TIMP-1 correlated with peak Troponin T (rho = 0.361, *p* = 0.042).	High circulating TIMP-1 plays an independent role in myocardial damage early post-MI
[40]	Plasma level of TIMPs by ELISA kit	Plasma level TIMP-1 and TIMP-2 were significantly decreased (*p* < 0.0001) in CAD (146.89 ± 73.12 ng/mL, 107.84 ± 45.07 ng/mL) when compared to control (327.44 ± 88.67 ng/mL, 278.85 ± 63.72 ng/mL)	Plasma TIMP decreased in CAD patients versus control
[41]	Measurement of gene expression of TIMP-3 from leucocyte, arteries, and EATs by quantitative RT-PCR	For living cases: TIMP3 expression levels of circulating leukocytes were not statistically different in the high plaque score group (1.47 ± 1.33, n = 48) compared to the low plaque score group (1.08 ± 0.70, n = 21) (*p* = 0.66) TIMP3 expression levels in peri-coronary EAT samples did not show any difference between the two groups For post-mortem: The expression levels of TIMP-3 in postmortem coronary arteries showed statistically significant differences between three groups (2.99 ± 2.62, n = 25 for CHD-nonP group, 1.46 ± 1.14, n = 26 for CHD-P group, and 1.13 ± 0.32, n = 4 for T-nonP, *p* = 0.02) The expression levels of TIMP3 in peri-coronary EAT between samples of CHD-P group (6.21 ± 6.46, n = 10), CHD-nonP group (4.53 ± 3.85, n = 10), and (*p* = 0.27 and T-nonP group (1.65 ± 1.86, n = 2) were not significant.	The reduction in TIMP3 expression is notable in advanced atherosclerotic plaques. Consequently, the elevated TIMP3 expression observed in the normal coronary arterial segments of individuals with CHD suggests that TIMP3 serves a protective function against atherosclerosis development at the molecular level within these arterial regions
[42]	IVUS-Allura Xper FD10 system Measurement of serum MMP-7, MMP-9, TIMP-2, TIMP-1: ELISA	MMP-7 and MMP-9 were highly expressed while in TIMP-2 was less expressed in fibroatheroma, which is the advance stage of atherosclerosis MMP-9 independently predicted necrotic size (β = 0.44, p < 0.05) Negative correlation between MMP-9 and TIMP-1 observed (r = −0.28, *p* = 0.04)	Circulating TIMP-2 was decreased in fibroatheroma type of atherosclerotic lesion
[43]	Measurement of serum TIMP2- ELISA	Serum TIMP-2 was significantly higher in patients with coronary artery stenosis [126.1(78)] vs. normal (80.6 ± 17) ng/dL. Circulating TIMP-2 level positively related to Gensini score after adjustment for age and sex (r = 0.298, *p* = 0.02) Circulating TIMP-2 level positively related to number of arteries with significant stenosis (J-T statistic = 3.01, *p* = 0.003)	Increased in TIMP-2 could be a potential biomarker to evaluate CAD

TIMP, tissue inhibitor matrix metalloproteinase; MMP, matrix metalloproteinase; TC, total cholesterol; TG, triglyceride; HDL-C, high-density lipoprotein cholesterol; LDL-C, low-density lipoprotein cholesterol; eGFR, estimated glomerular filtration rate; AMI, acute myocardial infarction; CAD, coronary artery disease; STEMI, ST elevation myocardial infarction; NSTEMI, non-ST elevation myocardial infarction; CVD, cardiovascular disease; EATs, epicardial adipose tissue; CHD-P, Coronary heart disease with advanced atherosclerotic plaque; CHD-nonP, Coronary heart disease with no atherosclerotic plaque; T-nonP, Non- cardiac trauma with non-atherosclerotic coronary artery/plaque; MRI, magnetic resonance imaging; ELISA, enzyme-linked immunosorbent assay; CMIA, chemiluminescent microparticle immunoassay; RT-PCR, reverse transcription polymerase chain reaction; CABG, coronary artery bypass graft surgery; CAG, coronary artery angiography; NOAC, novel oral anticoagulants; ACE-I, Angiotensin converting enzyme; AT-II-blocker, Angiotensin II blocker; RAASI, Renin-angiotensin-aldosterone system inhibitors; MACEs, major adverse cardiovascular events; ARBs, Angiotensin receptor blockers; PBMCs, peripheral blood mononuclear cells; SPECT, Single-Photon Emission Computed Tomograph; ASA, Acetylsalicylic acid; IQR, interquartile range; IVUS, intravascular ultrasound; LVEF, left ventricular ejection fraction; HF, heart failure.

## Data Availability

This study’s original contributions are documented within the article. For further inquiries, please contact the corresponding author directly.

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
