# Peer review of "Unveiling TIMPs: A Systematic Review of Their Role as Biomarkers in Atherosclerosis and Coronary Artery Disease"

_diseases, 2024, doi:10.3390/diseases12080177_

Round 1

Reviewer 1 Report

Comments and Suggestions for Authors

Some type is still in draft format (re Table 1 title).

Abstract could be more informative, e.g. that clinical studies involving established CAD were considered. An expression of confidence (based on quality of evidence) when reporting conclusions would be expected.

Introduction could better identify gaps in knowledge to be addressed by systematic review and be clearer on nature of studies/data (experimental vs clinical) as well as clearer on the sources of TIMP expression measurement, e.g. tissue sample vs circulating. The concept of biomarkers could be explored in more depth. A hypothesis for the relationship between TIMP, MMP and atherosclerosis progression could be more clearly and rigorously stated. 

The population of interest is probably more meaningfully restricted to CAD patients (as opposed to CAD and non-CAD patients). CAD might have been included as a key word re search strategy.

Why was the year range 2014-2024 selected? The Discussion for instance refers to research spanning over two decades. 

Inclusion criteria could include the requirement for peer-review re original articles/primary sources.

The respective roles of the NOS score and JBI tool in assessing the risk of bias and evaluation of study quality could be made much clearer. Table 2 features NOS scores but JBI outcomes are indicated in section 2.4 (and no JBI outcomes are shown)?

PRISMA needs to state 48 (i.e. 63-15) reports assessed for eligibility. Why were those 15 reports not retrieved out of the 63 sought for retrieval?

The PRISMA, data extraction and risk of bias score tables may be more appropriate under the Results section.

There would be expected to be some narrative reporting (e.g. re trial design, measurements, analysis, population and major outcomes) of each included, individual study re Results (in addition to a data extraction table).

Re Table 1 continued, it is not clear (in the present format) which study outcomes are being reported and the detail of what is being reported (e.g. circulating or tissue sample TIMP levels). Table could additionally indicate whether all reported outcomes were statistically significant or not; and also include the range of any patient medications where appropriate/available.

Overall the answer to the systematic review question remains inconclusive and ambiguous. In this regard, the systematic review would have benefited from a meta-analysis and the Discussion could have made use of e.g. bias/quality assessments and relative strengths of trial designs in considering the individual contributions of the studies (all studies seem to have been equally weighted). More critical insight could have been applied to help provide explanations as to why studies may not agree. Furthermore, more consideration might have been given to the potentially confounding effects on interpretation of both the diversity in atherosclerosis progression/cardiovascular outcomes between studies and any use of patient medications.

Conclusion is a little vague with no real sense of consensus or integration conveyed (only studies in support of TIMPs as biomarkers are highlighted) and no level of confidence expressed (based on strength/quality of the evidence) to accompany statements that either support or contradict the formulated systematic review question. More clearly delineated gaps in knowledge could have been identified re future research.

Comments on the Quality of English Language

Needs some moderation, including title.

Reviewer 2 Report

Comments and Suggestions for Authors

Revision Manuscript Aminuddin

Manuscript Number: diseases-3091001

The manuscript “Are TIMPs Emerging Biomarker of Coronary Artery Disease? A systematic review” by Aminuddin and colleagues summarizes the research findings on the specific role of TIMPs in the complex landscape of atherosclerosis. This systematic review was conducted to explore the emerging role of TIMPs as biomarkers in atherosclerosis and to highlight their complex involvement in vascular remodeling, inflammation and plaque stability. Overall, this is a well-written review that is informative for the reader. It gives the impression of a well thought out and carefully executed work. In particular, the structure of the entire manuscript and the description of the individual parts are very clear and comprehensible. However, there are some points that should be addressed by the authors to further improve the manuscript and help the reader to better understand this review.

Major comments:

·         Pages 4 - 9, Table 1: The table was difficult to read due to its formatting. There are too many columns to cleanly display on a DINA4 (US 8.5” x 11”) page. The authors seem to have split the table into two parts without a clear system to show which rows matched to the first table. This made reading the table more time-consuming. It would be better to choose an alternative presentation, or at least label the second table with the appropriate literature sources. It would be better to give each literature source its own table, rather than split one large table into two parts. This would also resolve problems regarding cells split between pages, which adds further confusion.

·         Page 9, line 159 – 163: The Newcastle-Ottawa Scale (NOS) was used to assess the risk of bias for each study. How is the NOS related to Joanna Briggs Institute (JBI) Critical Appraisal tools? Which JBI checklists were used for the case-control studies and the cohort/cross-sectional studies? NOS and JBI tools serve similar overarching purposes in terms of quality assessment, but they are designed for different types of studies and use different methodologies. 2-3 sentences of further explanation would be helpful here.

·         Page 10, line 181: “Besides, there were 80% (n=8) studies that were cohort studies, 10% (n=1) case-control studies and 10% (n=1) cross-sectional study.” In Table 2 and 3, the study types are case-control studies (n=2) (Table 3), cohort studies (n=5) (Table 2) and cross-sec studies (n=3) (Table 2). How is this to be understood? Why does the classification of the individual studies change here?

·         Page 11, line 189: "Among these 8 studies reviewed, TIMP 1 was...". Which 8 studies are meant here exactly?

Minor comments:

·         On page 2, line 81: “…role in the development of (12,13) an aneurysm." The literature reference would fit better at the end of the sentence.  

·         The methodology section is very nicely and clearly written, and the flow chart illustrates the process of selecting a screening based on the PRISMA guidelines very well. However, I personally feel that the text and the diagram could work better together. An example would be "Reports excluded due to not fulfilling all the inclusion criteria" (Fig. 1). What are the inclusion criteria? The criteria do not have to be listed in the figure, but a small indication of where exactly in the text (2.1, 2.2, 2.3 or 3.1) they can be found would make it easier for the reader to understand the individual steps.

·         Studies included in review (n=10)/Reports of included studies. Here it is not entirely clear where exactly the difference between the two points lies.

·         Page 9, line 145 – 158: The text with the abbreviations should be checked for consistency before resubmission. An example would be in line 149 "...apolipoproteins B100 TC, total cholesterol;...", ...apolipoproteins B100; TC, total cholesterol;.... A consistent style would make it easier for the reader to follow.

Round 2

Reviewer 1 Report

Comments and Suggestions for Authors

Many thanks for addressing and incorporating this reviewer's comments.